

# Differences in cervical sagittal parameters and muscular function among subjects with different cervical spine alignments: a surface electromyography-based cross-sectional study

Dian Wang[1], Shuanghe Liu[2], Yibo Liu[2] and Zheng Zeng[2]

[1] Department of Orthopaedic Surgery, Beijing Anzhen Hospital, Capital Medical University, Beijing, China
[2] Department of Orthopaedic Surgery, Beijing Tiantan Hospital, Capital Medical University, Beijing, China

Corresponding author
Zheng Zeng, cowandmilk@163.com

## ABSTRACT

**Background:** We analyzed cervical sagittal parameters and muscular function in different cervical kyphosis types.

**Methods:** This cross-sectional study enrolled subjects with cervical spine lordosis (cervical curvature < −4°) or degenerative cervical kyphosis (cervical curvature > 4°), including C-, S-, and R-type kyphosis. We recorded patients' general information (gender, age, body mass index), visual analog scale (VAS) scores, and the Neck Disability Index (NDI). Cervical sagittal parameters including C2–C7 Cobb angle (Cobb), T1 slope (T1S), C2–C7 sagittal vertical axis (SVA), spino-cranial angle (SCA), range of motion (ROM), and muscular function (flexion-relaxation ratio (FRR) and co-contraction ratio (CCR) of neck/shoulder muscles on surface electromyography). Differences in cervical sagittal parameters and muscular function in subjects with different cervical spine alignments, and correlations between VAS scores, NDI, cervical sagittal parameters, and muscular function indices were statistically analyzed.

**Results:** The FRR of the splenius capitis (SPL), upper trapezius (UTr), and sternocleidomastoid (SCM) were higher in subjects with cervical lordosis than in subjects with cervical kyphosis. $FRR_{SPL}$ was higher in subjects with C-type kyphosis than in subjects with R- and S-type kyphosis ($P < 0.05$), and was correlated with VAS scores, Cobb angle, T1S, and SVA. $FRR_{UTr}$ was correlated with NDI, SCA, T1S, and SVA. $FRR_{SCM}$ was correlated with VAS scores and Cobb angle. CCR was correlated with SCA and SVA.

**Conclusion:** Cervical sagittal parameters differed among different cervical kyphosis types. FRRs and CCRs were significantly worse in R-type kyphosis than other kyphosis types. Cervical muscular functions were correlated with cervical sagittal parameters and morphological alignment.

## INTRODUCTION

Due to its greater range of motion and the burden imposed by the head, the cervical spine is more susceptible than the thoracic or lumbar spine to various pathological changes, including degenerative alterations and malalignment (*Scheer, Lau & Ames, 2021*). *Ohara et al. (2006)* classified cervical spine malalignment into three types: straight, sigmoid, and global cervical kyphosis, with each type accounting for approximately one-third of the total number of cases. *Ao et al. (2019)* found that 38.3% of the normal population exhibited cervical spine kyphosis, and the incidence of cervical malalignment was higher in young people than in older people.

Cervical malalignment not only increases the tension on the spinal cord, leading to severe dysfunction and reduced health-related quality of life (*Ao et al., 2019*; *Ruangchainikom et al., 2014*; *Scheer, Lau & Ames, 2021*), but also affects the overall balance of the spine. The thoracic spine, lumbar spine, and pelvis then undergo compensatory changes to maintain a horizontal gaze (*Patwardhan et al., 2018*; *Yu et al., 2015*). Similar to the straight type of cervical malalignment, cervical kyphosis can also cause headache, neck and shoulder pain, and masticatory dysfunction, and significantly increases energy expenditure (*Been, Shefi & Soudack, 2017*).

In recent years, an increasing number of studies have focused on the influence of neck and shoulder muscular function on cervical curvature and sagittal balance, and found that cervical muscle dysfunction is related to cervical spine deformity and postoperative sagittal imbalance (*Tamai et al., 2019*; *Passias et al., 2018*; *Wang et al., 2020*). *Abelin-Genevois (2021)* reported that normal spinal alignment is based on the balance between neurally modulated muscle responses and external forces (gravity). *Xu et al. (2023)* showed that paraspinal muscle atrophy is associated with pain, spondylolisthesis, and facet joint degeneration. *Wang et al. (2020)* found that cervical extensor muscle function differed between subjects with cervical malalignment and those with cervical lordosis, and demonstrated that cervical muscular function was correlated with cervical sagittal parameters. Compared with morphological muscle-evaluation methods, surface electromyography (sEMG) is a dynamic and function-based method that offers advantages in the evaluation of neck and shoulder muscular function (*Wang et al., 2020*). *Yu et al. (2013)* reported that cervical sagittal parameters differed with cervical alignment morphology (lordosis, straight, sigmoid kyphosis, and global kyphosis). However, the authors of the above study did not evaluate neck and shoulder muscular function, and the differences in muscular function among subjects with different types of cervical malalignment remain unknown.

Therefore, we designed a cross-sectional study of subjects with cervical lordosis and those with different types of cervical malalignment. We collected the general information, quality of life scores, cervical spine sagittal parameters, and muscular function indices of the subjects. The objectives of this study were two-fold: (1) to evaluate differences in quality of life scores, cervical spine sagittal parameters, and muscular function indices among subjects with different cervical spine alignments, namely, cervical lordosis, and C-type, S-type, or R-type cervical kyphosis, and (2) to analyze the correlations of quality of
life scores, cervical spine imaging parameters, and muscle function indices with cervical spine alignment. We used sEMG to explore the influence of muscular factors on the biomechanical patterns and sagittal balance of the cervical spine, and to provide a reference for symptom assessment and treatment planning of patients with cervical kyphosis.

# MATERIALS AND METHODS

## Inclusion and exclusion criteria

The inclusion criteria for this study were as follows: (1) age ≥ 18 years, (2) $18 \leq$ body mass index (BMI) $\leq 35$ kg/m$^2$; (3) lateral cervical dynamic radiographs (DR) showing a C2–C7 Cobb angle of $<-4°$ (defined as cervical lordosis) or $>4°$ (defined as cervical kyphosis) (*Ruangchainikom et al., 2014*; *Yoon et al., 2018*), and (4) written informed consent from the subjects.

The following exclusion criteria were applied: (1) patients with cognitive impairment who were unable to cooperate with the examination, (2) patients with a neuromuscular disorder, sarcopenia, or other diseases that affect the assessment of muscle function (*Chiou-Tan, 2022*; *Ding et al., 2019*), (3) patients with unstable vital signs or cachexia, (4) patients with a visual analog scale (VAS) score of $>3$ for neck pain (*Falla et al., 2007*), (5) Patients with cervical spondylotic myelopathy or cervical instability and (6) patients with incomplete data on quality of life scores, cervical sagittal imaging parameters, or muscular function indices.

## Grouping criteria

The alignment of the cervical spine was determined according to the relative position of the center of gravity of each cervical vertebra on lateral cervical DR, and a line was drawn connecting the centers of gravity of the C2 and C7 vertebrae (termed the AB line) (*Ohara et al., 2006*). C-type kyphosis was identified when the centers of gravity of the cervical vertebrae were distributed behind the AB line, and the maximum distance between the AB line and at least one center of gravity was ≥2 mm. S-type kyphosis was diagnosed when the centers of gravity of the vertebrae were distributed on both sides of the AB line, and the maximum distance between the AB line and the most posterior vertebra was ≥2 mm; in addition, the most posterior vertebra was below the C5 segment. R-type kyphosis was detected when the vertebral centers of gravity were distributed on both sides of the AB line, and the maximum distance between the AB line and the most posterior vertebra was ≥2 mm; in addition, the most posterior vertebra was above the C5 segment (*Toyama & Kamata, 1997*) (Fig. 1).

## Sample size calculation

The required sample size was calculated using the following formula: $n = (\mu_\alpha + \mu_\beta)^2 \sigma^2 / \delta^2$, where $\alpha$ was set as 0.05, and $\beta$ was set as 0.01. According to the $\mu$ table, the values of $\mu_\alpha$ and $\mu_\beta$ were 1.96 and 1.2816, respectively. The values of $\sigma$ (5.57) and $\delta$ (3.30) were determined in a pre-experiment (which included 20 patients with C-type kyphosis of the lower cervical spine). The sample size for each group was calculated as 30 patients, and then expanded by 20% to

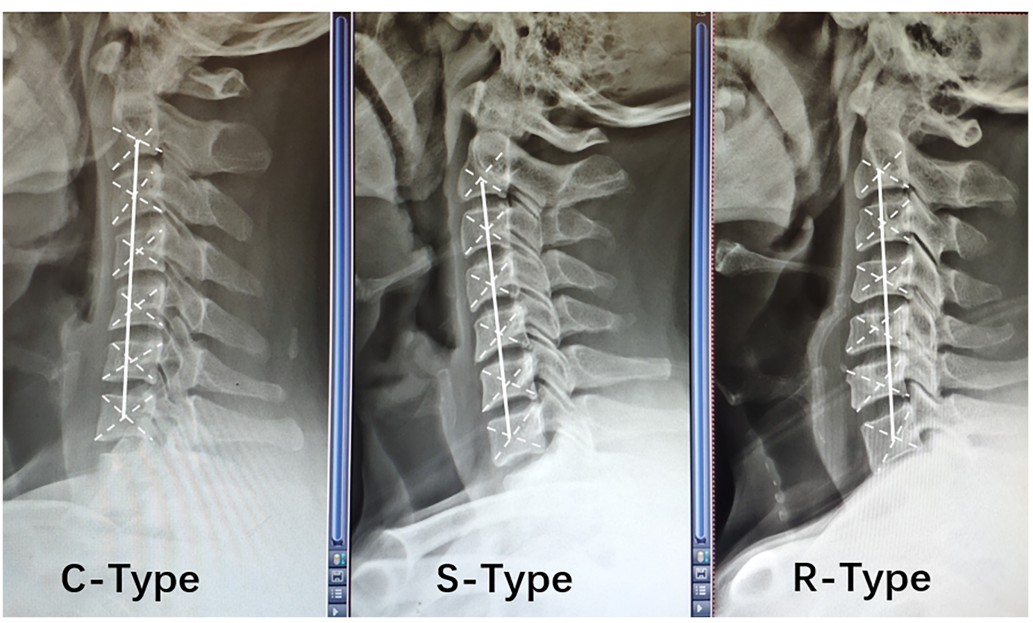

**Figure 1 Relative positions of the center of gravity of each vertebral body in different types of cervical spine kyphosis.**

36% patients. Finally, 70 patients with cervical lordosis, 44 patients with C-type kyphosis, 39 patients with S-type kyphosis, and 41 patients with R-type kyphosis were enrolled.

## Research methods

The ethics committee of our hospital (Beijing Tiantan Hospital, Capital Medical University) approved this study according to the human subject protection programs and procedures (Ethics review approval number: KY2021-254-03). Written informed consent was received from participants of our study.

A cross-sectional study was designed. After the subjects were enrolled, we performed data collection and analysis as shown in the flow chart in Fig. 2. The general information of the subjects was collected, including age, gender, and BMI. The patients' quality of life scores, including VAS scores for neck pain and the Neck Disability Index (NDI), were determined. Lateral and dynamic radiographs of the cervical spine were taken to measure radiographic parameters, including C2–C7 Cobb angle (Cobb), spino-cranial angle (SCA), T1 slope (T1S), C2–C7 cervical sagittal vertical axis (SVA), and range of motion (ROM) (*Le Huec et al., 2019*). All subjects underwent sEMG-based assessments to evaluate muscular functions, including flexion-relaxation ratio (FRR) of the splenius capitis (SPL), upper trapezius (UTr), and sternocleidomastoid (SCM), and the co-contraction ratio (CCR).

## Evaluation criteria

### Quality of life score

The degree of neck pain was evaluated using VAS scores (*McCormack, Horne & Sheather, 1988*). The degree of disability of the subjects due to neck pain was evaluated using the NDI (*Vernon & Mior, 1991*).

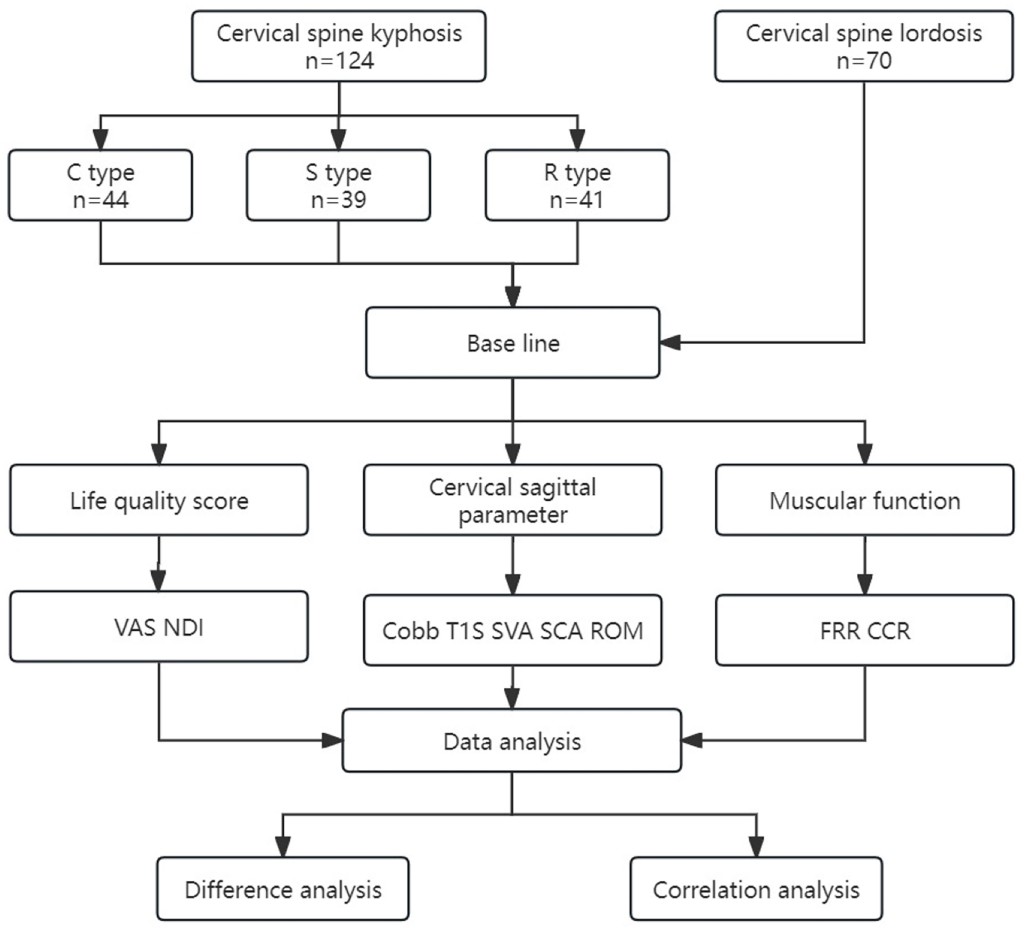

**Figure 2 Study design.** VAS, visual analog scale; NDI, Neck Disability Index; Cobb, C2–C7 Cobb angle; T1S, T1 slope; SVA, C2–C7 sagittal vertical axis; SCA, spino-cranial angle; ROM, range of motion; FRR, flexion-relaxation ratio; CCR, co-contraction ratio.   

## Imaging parameters

Lateral cervical radiographs and cervical flexion and extension radiographs (cervical spine dynamic radiographs) were obtained for each patient. The Cobb angle, SCA, T1S, and C2–C7 SVA were measured (*Ling et al., 2018*). With the subjects in the upright position, the lower part of the cassette was kept in close contact with the lateral border of the shoulder; with the shoulders naturally drooped, the subjects actively performed the maximum flexion and maximum extension of the neck. Cervical spine flexion and extension radiographs were used to evaluate the presence of cervical dynamic instability and measure the ROM of the neck (*Li et al., 2018*). Subjects with cervical spine instability were excluded, according to the selection criteria.

## Muscular function

Bioelectrical signals from the neck and shoulder muscles were collected using an sEMG device (FlexComp Infiniti System, T7550; Thought Technology Ltd., Montreal, Canada), subjects took the standard sitting position, relaxed their upper limbs and drooped
naturally. The ambient temperature was 22–24 °C (71.6–75.2°F) and the relative humidity was 40% (*De Carvalho et al., 2024*). The SPL, UTr, and SCM muscles in the neck and shoulders were selected for measurement, as they are relatively superficial, isolated, and suitable for sEMG measurements (*Cheng, Lin & Wang, 2008*). The SPL collection point was located 2 cm lateral to the C4 spinous level (*Ding et al., 2019*; *Wang et al., 2020*); this is the most suitable point for the acquisition of SPL signals because of the lack of interference from the UTr EMG signals (*Dieterich et al., 2016*). UTr signals were collected at the midpoint of a line connecting the C7 spinous process and the acromion, and SCM signals were collected at the junction of the middle and lower thirds of a line connecting the sternal notch and the mastoid bone (*Wang et al., 2020*) (Fig. 3). The sEMG-based FRR and CCR are commonly used to evaluate muscle function (*Cheng et al., 2014*; *Ding et al., 2019*; *Nimbarte, Zreiqat & Chowdhury, 2014*; *Thelen, Schultz & Ashton-Miller, 2010*). The FRR reflects the ratio of the ability of a muscle to tense during active contraction and relax during passive stretching (*Wang et al., 2020*), which represents the elasticity of the muscle. A decrease in the FRR indicates muscle stiffness (*Nimbarte, Zreiqat & Chowdhury, 2014*). *De Carvalho et al. (2024)* reported that the FRR is associated with good test reliability. The CCR reflects the degree of simultaneous activation of active and antagonistic muscles during neck movement. An increase in the CCR will limit the neck ROM and improve the stability of the cervical spine (*Cheng, Lin & Wang, 2008*).

## Statistical analysis

The statistical software SPSS 24.0 (IBM, Armonk, NY, USA) was used for statistical analysis. The intraclass correlation coefficient (ICC) values ranged from 0.79 to 0.85 (>0.75). The data consistency was reliable. Differences in continuous variables among multiple groups were evaluated using analysis of variance. Pearson analysis was used to analyze correlations between continuous variables, and Spearman analysis was used to analyze correlations between discontinuous variables. The two-sided test with an α level of 0.05 was considered to indicate statistical significance.

## RESULTS

### General information

According to the inclusion and exclusion criteria, 124 subjects diagnosed with degenerative cervical kyphosis and 70 subjects with cervical lordosis who visited our orthopedic department between September 2022 and December 2023 were consecutively enrolled, including 44 patients with C-type kyphosis, 39 patients with S-type kyphosis, and 41 patients with R-type kyphosis. The lordosis group consisted of 32 men and 38 women, with an average age of 45.4 ± 17.2 years (range, 19–71 years) and an average BMI of 24.7 ± 3.3 kg/m$^2$ (range, 20.5–29.5 kg/m$^2$). The C-type kyphosis group included 21 men and 23 women, with an average age of 43.0 ± 15.3 years (range, 19–65 years) and an average BMI of 24.9 ± 2.9 kg/m$^2$ (range, 18.3–30.8 kg/m$^2$). The S-type kyphosis group contained 15 men and 24 women, with an average age of 47.2 ± 17.4 years (range, 19–77 years) and an average BMI of 23.8 ± 3.5 kg/m$^2$ (range, 18.4–30.3 kg/m$^2$). The R-type kyphosis group consisted of 18 men and 23 women, with an average age of 46.8 ± 17.3 years

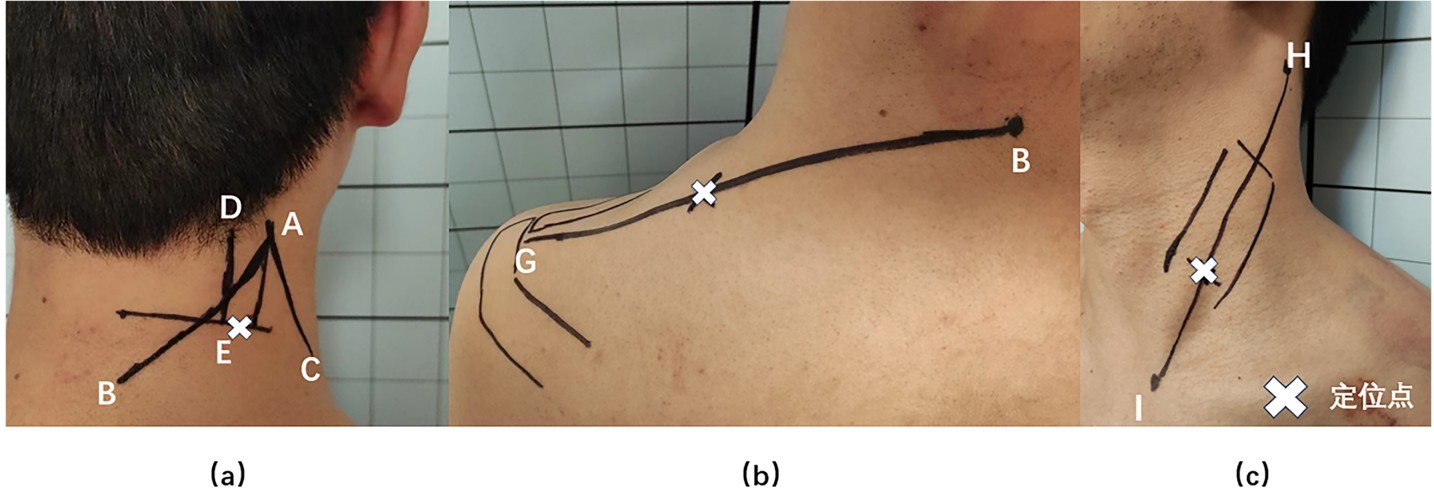

(a)  (b)  (c)

**Figure 3 Splenius capitis, upper trapezius, and sternocleidomastoid muscle surface anchor points.** (A) Splenius capitis body surface anchor points: line AB, the line between the apex of the auricle and the C7 spinous process; line AC, the posterior edge of the sternocleidomastoid muscle; line DE, the belly of the splenius capitis muscle; and an anchor point located outside point E. (B) Upper trapezius surface anchor points: line BG, line between the C7 spinous process and the acromion; the anchor point is at the midpoint of line BG. (C) Sternocleidomastoid body surface anchor points: line HI, line connecting the sternal notch and the mastoid bone; the fixation point was located at the junction of the middle and lower thirds of line HI.

(range, 19–77 years) and an average BMI of $25.2 \pm 4.1$ kg/cm$^2$ (range, 18.3–34.3 kg/cm$^2$). No significant differences in gender, age, and BMI were present among the above groups ($P > 0.05$).

## Differences in cervical sagittal parameters

The Cobb angle was $-12.80° \pm 5.71°$ in the cervical lordosis group, $17.82° \pm 7.78°$ in the C-type kyphosis group, $10.92° \pm 4.84°$ in the S-type kyphosis group, and $12.24° \pm 3.11°$ in the R-type kyphosis group. Multiple comparisons of means showed that the Cobb angle was significantly higher in the cervical kyphosis groups than in the cervical lordosis group ($P < 0.01$). Moreover, the Cobb angle in the C-type kyphosis group was significantly higher than those in the R-type and S-type kyphosis groups ($P < 0.01$), while no significant difference in the C2–C7 Cobb angle was found between the S-type and R-type kyphosis groups.

The SCA was $83.77° \pm 9.89°$ in the cervical lordosis group, $89.57° \pm 7.47°$ in the C-type kyphosis group, $81.10° \pm 6.82°$ in the S-type kyphosis group, and $81.51° \pm 5.95°$ in the R-type kyphosis group. Multiple comparisons showed that the SCA was significantly higher in the C-type kyphosis group than in the cervical lordosis, S-type kyphosis, and R-type kyphosis groups ($P < 0.01$), whereas the SCA did not significantly differ between the cervical lordosis, S-type kyphosis, and R-type kyphosis groups.

The T1S was $16.00° \pm 12.41°$ in the cervical lordosis group, $10.57° \pm 7.24°$ in the C-type kyphosis group, $12.67° \pm 8.64°$ in the S-type kyphosis group, and $7.02° \pm 5.08°$ in the R-type kyphosis group. Multiple comparisons showed that T1S was significantly higher in the cervical lordosis group than in the C-type and R-type kyphosis groups ($P < 0.01$), and the T1S in the S-type kyphosis group was significantly higher than that in the R-type

kyphosis group ($P = 0.047$). No significant difference in T1S was observed between the C-type and R-type kyphosis groups.

The SVA was $24.44 \pm 17.25$ mm in the cervical lordosis group, $20.05 \pm 15.35$ mm in the C-type kyphosis group, $20.31 \pm 17.47$ mm in the S-type kyphosis group, and $8.17 \pm 10.22$ mm in the R-type kyphosis group. The SVA was significantly lower in the R-type kyphosis group than in the other three groups ($P < 0.01$). No significant differences in SVA were detected among the remaining three groups.

The ROM was $45.14° \pm 11.02°$ in the cervical lordosis group, $44.98° \pm 6.93°$ in the C-type kyphosis group, $40.13° \pm 8.23°$ in the S-type kyphosis group, and $46.00° \pm 16.01°$ in the R-type kyphosis group. Multiple comparisons showed that the ROM was significantly lower in the S-type kyphosis group than in the cervical lordosis group ($P = 0.049$) and C-type kyphosis group ($P = 0.030$; Fig. 4).

### Differences in muscular function

The FRR of the SPL was $2.86 \pm 1.08$ in the cervical lordosis group, $2.05 \pm 0.66$ in the C-type kyphosis group, $1.86 \pm 0.83$ in the S-type kyphosis group, and $2.21 \pm 0.54$ in the R-type kyphosis group. Multiple comparisons showed that the $FRR_{SPL}$ was significantly higher in the cervical lordosis group than in the C-, S-, and R-type kyphosis groups ($P < 0.01$), but this ratio did not significantly differ among the three kyphosis groups.

The FRR of the UTr was $2.41 \pm 0.90$ in the cervical lordosis group, $2.62 \pm 1.16$ in the C-type kyphosis group, $2.16 \pm 1.08$ in the S-type kyphosis group, and $1.74 \pm 0.64$ in the R-type kyphosis group. Multiple comparisons showed that the $FRR_{UTr}$ was significantly higher in the cervical lordosis and C-type kyphosis groups than in the R- and S-type kyphosis groups ($P < 0.01$). No significant difference in this ratio was observed between the S-type and R-type kyphosis groups.

The FRR of the SCM was $3.35 \pm 1.32$ in the cervical lordosis group, $3.02 \pm 2.74$ in the C-type kyphosis group, $2.64 \pm 1.20$ in the S-type kyphosis group, and $2.52 \pm 0.80$ in the R-type kyphosis group. Multiple comparisons showed that the $FRR_{SCM}$ was significantly higher in the cervical lordosis group than in the S-type kyphosis group ($P = 0.032$) and R-type kyphosis group ($P < 0.01$). This ratio did not significantly differ among the three kyphosis groups.

The CCR was $0.77 \pm 0.22$ in the cervical lordosis group, $0.80 \pm 0.21$ in the C-type kyphosis group, $0.84 \pm 0.12$ in the S-type kyphosis group, and $0.65 \pm 0.16$ in the R-type kyphosis group. Multiple comparisons showed that the CCR was significantly higher in the cervical lordosis, C-type kyphosis, and S-type kyphosis groups than in the R-type kyphosis group ($P < 0.01$), but the CCR did not significantly differ among the cervical lordosis, C-type kyphosis, and S-type kyphosis groups (Figs. 5 and 6).

### Correlation analysis of general condition, quality of life scores, sagittal parameters, and muscular functions

Pearson correlation analysis (Table 1) showed that gender was correlated with ROM ($r = 0.215$, $P < 0.01$). Age was correlated with SCA ($r = -0.253$, $P < 0.01$), SVA ($r = 0.889$, $P < 0.01$), and ROM ($r = -0.156$, $P < 0.05$). BMI was correlated with ROM ($r = 0.214$,

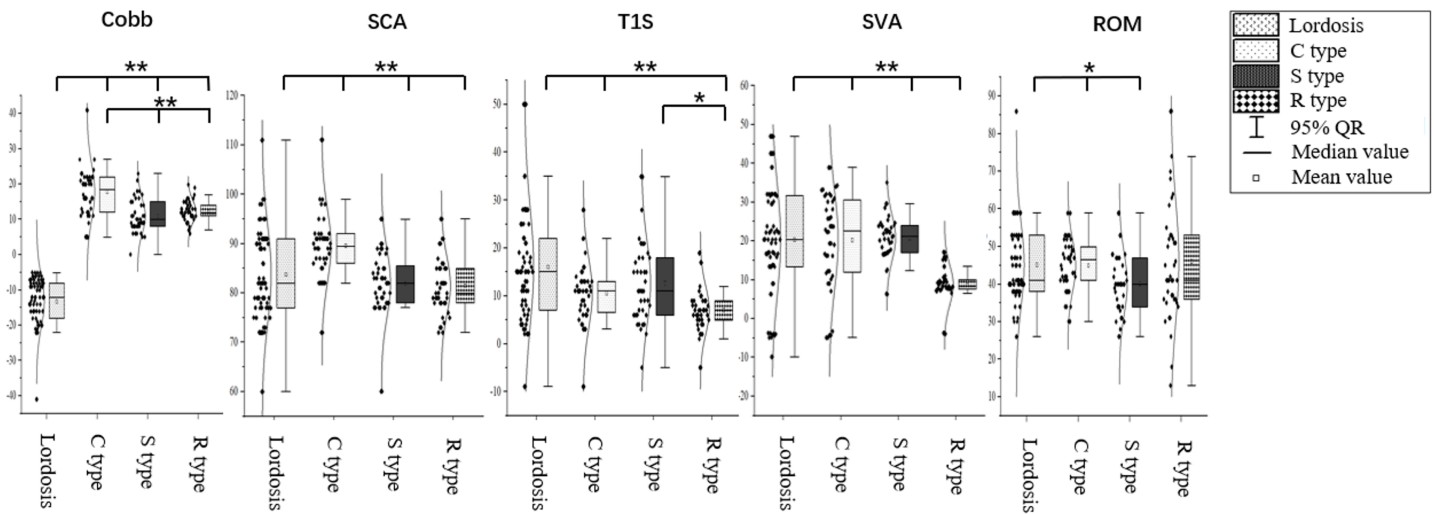

**Figure 4 Differences in cervical sagittal parameters in subjects with different morphological alignments of the cervical spine.** C2–C7 Cobb angle, Cobb; SCA, spino-cranial angle; T1S, T1 slope; SVA, C2–C7 sagittal vertical axis; ROM, range of motion. *$P < 0.05$; **$P < 0.01$.

$P < 0.01$). VAS scores were positively correlated with NDI ($r = 0.547$, $P < 0.01$) and C2–C7 Cobb angle ($r = 0.516$, $P < 0.01$), and negatively correlated with SCA ($r = -0.155$, $P < 0.05$), ROM ($r = -0.161$, $P < 0.05$), $FRR_{SPL}$ ($r = -0.197$, $P < 0.01$), and $FRR_{SCM}$ ($r = -0.176$, $P < 0.05$). NDI was positively correlated with C2–C7 Cobb angle ($r = 0.362$, $P < 0.01$) and $FRR_{UTr}$ ($r = 0.186$, $P < 0.01$). C2–C7 Cobb angle was negatively correlated with T1S ($r = -0.236$, $P < 0.01$), SVA ($r = -0.228$, $P < 0.01$), $FRR_{SPL}$ ($r = -0.401$, $P < 0.01$), and $FRR_{SCM}$ ($r = -0.159$, $P < 0.05$). SCA was negatively correlated with T1S ($r = -0.236$, $P < 0.01$), SVA ($r = -0.179$, $P < 0.05$), and ROM ($r = -0.171$, $P < 0.05$). SCA was positively correlated with $FRR_{UTr}$ ($r = 0.217$, $P < 0.01$) and CCR ($r = 0.225$, $P < 0.01$). T1S was positively correlated with SVA ($r = 0.239$, $P < 0.01$), $FRR_{SPL}$ ($r = 0.248$, $P < 0.01$), $FRR_{UTr}$ ($r = 0.150$, $P < 0.05$). SVA was negatively correlated with $FRR_{SPL}$ ($r = -0.060$, $P < 0.05$), and positively correlated with the $FRR_{UTr}$ ($r = 0.092$, $P < 0.05$) and CCR ($r = 0.570$, $P < 0.01$). The $FRR_{SPL}$ was positively correlated with $FRR_{UTr}$ ($r = 0.467$, $P < 0.01$) and CCR ($r = 0.523$, $P < 0.01$). The $FRR_{UTr}$ was positively correlated with CCR ($r = 0.591$, $P < 0.01$).

Spearman correlation analysis (Table 1) showed that gender (0 = female, 1 = male) was positively correlated with ROM ($r = 0.215$, $P < 0.01$). Cervical spine alignment (1 = cervical lordosis, 2 = C-type kyphosis, 3 = S-type kyphosis, 4 = R-type kyphosis) was positively correlated with VAS score ($r = 0.477$, $P < 0.01$), NDI ($r = 0.154$, $P < 0.05$), and C2–C7 Cobb angle ($r = 0.635$, $P < 0.01$), and negatively correlated with SVA ($r = -0.326$, $P < 0.01$), T1S ($r = -0.308$, $P < 0.01$), $FRR_{SPL}$ ($r = -0.315$, $P < 0.01$), $FRR_{UTr}$ ($r = -0.256$, $P < 0.01$), and CCR ($r = -0.242$, $P < 0.01$).

## DISCUSSION

In this study, we used sEMG to evaluate differences in muscular function among subjects with different cervical spine alignments, and assessed the correlation between muscular function and cervical sagittal parameters. *Vasavada, Li & Delp (1998)* found that cervical

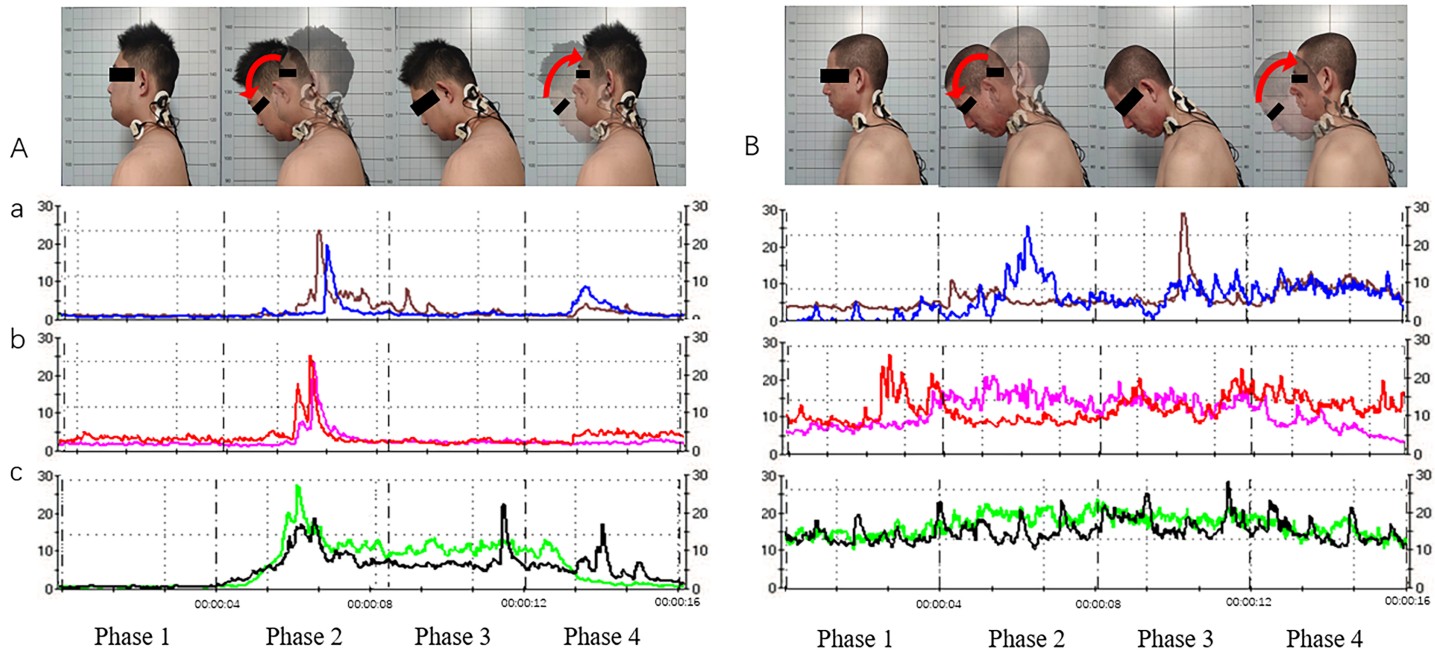

**Figure 5 Acquisition of neck and shoulder muscle signals *via* surface electromyography (sEMG; for the cervical extensor muscles as an example).** (A) Movement and muscle electrical signals of subjects with cervical lordosis. (B) Movement and muscle electrical signals of subjects with cervical kyphosis. a, Splenius capitis sEMG signals; b, upper trapezius sEMG signals; c, sternocleidomastoid sEMG signals. Phase 1, neutral position; phase 2, neutral position with gradual neck flexion to flexion position; phase 3, maintain the flexion position; phase 4, gradual extension from flexion position to neutral position.

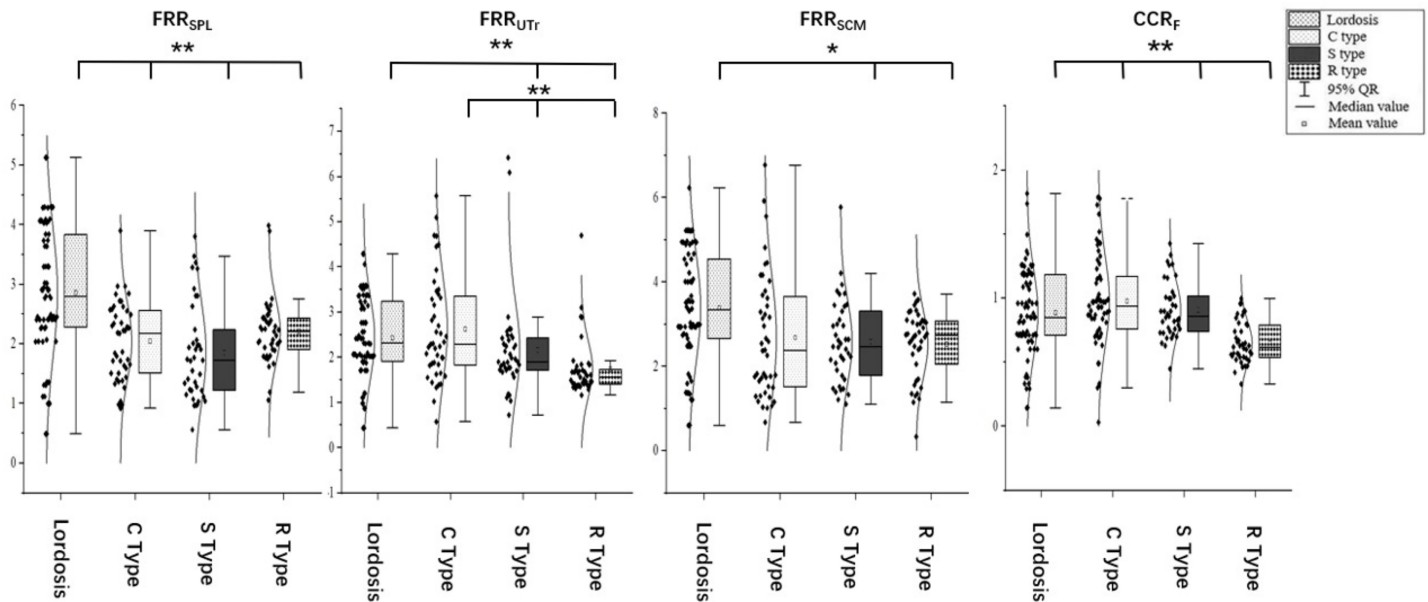

**Figure 6 Differences in muscular function in subjects with different cervical spinal alignments.** $FRR_{SPL}$, flexion-relaxation ratio of the splenius capitis; $FRR_{UTr}$, flexion-relaxation ratio of the upper trapezius; $FRR_{SCM}$, flexion-relaxation ratio of the sternocleidomastoid; CCR, co-contraction ratio. *$P < 0.05$; **$P < 0.01$.

**Table 1  Correlation analysis of general condition, quality of life scores, imaging parameters, and muscle function indices.**

| | Gender | Age | BMI | VAS | NDI | Cobb | SCA | T1S | SVA | ROM | Morphology | FRR$_{SPL}$ | FRR$_{UTr}$ | FRR$_{SCM}$ | CCR |
|---|---|---|---|---|---|---|---|---|---|---|---|---|---|---|---|
| Gender | 1.000 | | | | | | | | | | | | | | |
| Age | -0.12 | 1.000 | | | | | | | | | | | | | |
| BMI | 0.087 | -0.029 | 1.000 | | | | | | | | | | | | |
| VAS | -0.076 | 0.062 | 0.013 | 1.000 | | | | | | | | | | | |
| NDI | -0.037 | 0.080 | -0.008 | 0.547** | 1.000 | | | | | | | | | | |
| Cobb | -0.10 | -0.042 | 0.002 | 0.516** | 0.362** | 1.000 | | | | | | | | | |
| SCA | -0.063 | -0.253** | 0.005 | -0.155* | -0.099 | 0.057 | 1.000 | | | | | | | | |
| T1S | -0.027 | 0.137 | -0.020 | 0.102 | 0.137 | -0.236** | -0.439** | 1.000 | | | | | | | |
| SVA | -0.013 | 0.889** | -0.027 | -0.073 | 0.008 | -0.228** | -0.179* | 0.239** | 1.000 | | | | | | |
| ROM | 0.215** | -0.156* | 0.214** | -0.161* | 0.008 | -0.041 | -0.171* | -0.051 | -0.117 | 1.000 | | | | | |
| Morphology | -0.032 | 0.033 | -0.008 | 0.477** | 0.154* | 0.635** | -0.133 | -0.308** | -0.318** | -0.065 | 1.000 | | | | |
| FRR$_{SPL}$ | -0.008 | -0.127 | -0.001 | -0.197** | -0.055 | -0.401** | 0.118 | 0.248** | -0.060* | 0.115 | -0.315** | 1.000 | | | |
| FRR$_{UTr}$ | 0.005 | -0.039 | 0.033 | -0.031 | 0.186** | -0.083 | 0.226** | 0.150* | 0.092* | 0.030 | -0.256** | 0.467** | 1.000 | | |
| FRR$_{SCM}$ | 0.051 | -0.020 | 0.079 | -0.176* | 0.131 | -0.159* | -0.100 | 0.073 | 0.054 | 0.091 | -0.242** | -0.003 | 0.115 | 1.000 | |
| CCR | -0.036 | -0.113 | -0.019 | 0.044 | 0.118 | 0.043 | 0.225** | 0.106 | -0.015* | 0.059 | -0.138 | 0.523** | 0.591** | 0.066 | 1.000 |

**Notes:**
BMI, body mass index; VAS, visual analog scale; NDI, Neck Disability Index; Cobb, C2–C7 Cobb angle; SCA, spino-cranial angle; T1S, T1 slope; SVA, C2–C7 sagittal vertical axis; ROM, range of motion; Cervical spine alignment (1 = cervical lordosis, 2 = C-type kyphosis, 3 = S-type kyphosis, 4 = R-type kyphosis); FRR$_{SPL}$, flexion-relaxation ratio of splenius capitis; FRR$_{UTr}$, flexion-relaxation ratio of upper trapezius; FRR$_{SCM}$, flexion-relaxation ratio of sternocleidomastoid; CCR, co-contraction ratio.
* $P < 0.05$
** $P < 0.01$.

extensor muscles, including the splenius capitis and upper trapezius, play a dynamic stabilizing role during cervical spine movement. *Passias et al. (2018)* found that muscle atrophy and fatty infiltration were strongly correlated with cervical spine degeneration. *Ding et al. (2019)* and *Wang et al. (2020)* also found that cervical extensor muscles were responsible for cervical tension and stiffness in patients with cervical spine malalignment.

*Pialasse et al. (2009)* defined elastic muscles as those with an FRR > 2.5, and reported that an FRR < 2.5 represented poor muscle function and muscle stiffness. The results of this study showed that the FRRs of the neck and shoulder muscles of subjects with cervical lordosis were significantly higher than those of subjects with cervical kyphosis, suggesting that muscular function and muscle elasticity were better among subjects with cervical lordosis than among subjects with cervical kyphosis. Consistent with our findings, *Tamai et al. (2019)* and *Yoon et al. (2018)* reported that cervical muscular functions were worse in patients with cervical sagittal imbalance. *Ding et al. (2019)* found that cervical extensor muscles were responsible for tension in patients with cervical malalignment. Compared to previous studies, our study not only quantitatively assessed the flexion and extension muscular functions in patients with cervical lordosis and kyphosis, but also classified cervical kyphosis morphologies. According to the alignment of the cervical spine, cervical spine kyphosis is divided into the C, R, and S types (*Ohara et al., 2006*). Differences in neck and shoulder muscular functions were evaluated in subjects with different morphological types of cervical kyphosis.

This study found that there was no significant difference in the FRR of the SPL among subjects with different types of cervical kyphosis, suggesting that there was no significant difference in the relaxation and contraction function of the SPL among these subjects. We believe that this phenomenon is determined by anatomical characteristics; the SPL originates from the T3–T5 spinous processes and inserts into the C1–C2 transverse processes (*Brumpt et al., 2021*). The distance between the T3–T5 spinous processes and the C1–C2 transverse processes differed between subjects with cervical lordosis and kyphosis, which caused traction on the SPL. Compared to subjects with cervical lordosis, the cervical kyphosis subjects showed no difference in the function of the SPL; this is because the SPL crosses all possible vertices of the kyphosis.

The FRR of the UTr was higher in the cervical lordosis and C-type kyphosis groups than in the S-type and R-type kyphosis groups. The UTr originates from the occipital bone, and C1–C6 ligaments and spinous processes, and inserts into the posterior edge of the distal 1/3rd of the clavicle (*Camargo & Neumann, 2019*). The position of the apex of the kyphosis affects the distance between the origin and insertion of the UTr, which can explain the difference in muscular functions between subjects with global kyphosis and segmental kyphosis. *Lee et al. (2019)* found that there exists an interaction between cervical curvature and thoracic kyphosis. *Larsson, Søgaard & Rosendal (2007)* reported that musculoskeletal disorders of the neck and shoulders affect each other through the UTr. Considering the above findings along with the findings of this study, we believe that the sagittal imbalance of the cervical spine will affect the position of the origin of the UTr, thereby altering the distance between the origin and insertion of the UTr, and disturbing muscular

flexion-relaxation function. Our findings also indicate that cervical sagittal imbalance will potentially impact the balance of the shoulder joint.

In addition, this study found that the flexion-relaxation function of the SCM in the cervical lordosis group differed from that in the cervical segmental kyphosis groups (S type and R type). This phenomenon cannot be explained by anatomical factors since the SCM originates from the sternal notch and the inner one-third of the clavicle, and terminates in the mastoid process of the temporal bone. Correlation analysis revealed a positive correlation in the muscular functions of the SCM and UTr. Therefore, we suggest that even though segmental kyphosis does not affect the position of the SCM, due to the co-contraction between the SCM and UTr, the effects of the change in the alignment of the cervical spine on the UTr are transmitted to the SCM, thereby changing the relaxation and contraction functions of the SCM.

This study not only evaluated the elasticity of neck and shoulder muscles, but also calculated the contraction function and energy consumption during cervical movement. We found that the energy consumption during cervical flexion was significantly higher in the lordosis, C-type kyphosis, and S-type kyphosis groups than in the R-type kyphosis group. *Pialasse et al. (2010)* reported that the gravity load of the head on the normal cervical curvature can be transmitted from the musculature to the skeletal-ligament tissues, avoiding continuous tension of the muscles. *Wang et al. (2017)* determined that the gravity load of the head was significantly lower in subjects with cervical spine lordosis than in subjects with cervical malalignment. In this study, only subjects with R-type kyphosis had a significantly lower CCR than that of the cervical lordosis group, C-type and S-type kyphosis group, which indicates lower energy consumption. We found that the C2–C7 SVA was significantly lower among subjects with R-type kyphosis than among subjects with cervical spine lordosis, C-type kyphosis, and S-type kyphosis; this result represents the relatively low, forward head posture (FHP) seen in the R-type kyphosis group. *Bokaee et al. (2017)* used musculoskeletal ultrasonography to evaluate the morphology of neck and shoulder muscles, and found that compared with subjects with a normal head position, subjects with a FHP showed significantly increased SCM thickness; however, the morphology of the cervical extensor muscles did not differ between the two groups. These changes disrupted the overall balance of the neck and shoulder muscles. In a study of patients with FHP who underwent non-surgical treatment, *Choi (2021)* found that the manual release of the posterior neck muscles could significantly improve appearance and relieve pain, but its specific effects are unknown since the authors did not evaluate muscle function. Due to a lack of qualitative and quantitative assessment of muscle function, the effect of the cervical muscles on FHP remains unknown. Our study used sEMG to qualitatively and quantitatively evaluate the flexion and extension functions of muscles. Correlation analysis showed that an increase in FHP was positively correlated with extensor muscle function. Multivariate regression analysis also showed that neck pain, FHP, and elasticity of the SPL and UTr were influencing factors for the co-contraction function of the cervical spine. During neck flexion, the FHP is increased, and the center of gravity of the head is moved forward, increasing the risk of instability. To maintain cervical stability, the output of the extensor muscles, the proportion of antagonistic growth, as well

as the energy consumption of cervical movement are all increased. Although an increased CCR can improve the overall stability during cervical spine movement by forming a muscular collar, the cost is an increase in the overall energy consumption of neck movements.

T1S and C2–C7 SVA are the most important cervical sagittal balance parameters (*Le Huec et al., 2019*). Our study found T1S and C2–C7 SVA are positively correlated with cervical extensor function (SPL and UTr) as well as with muscular co-contraction functions. *Wang et al. (2020)* found that with increase in the T1 tilt angle, the elasticity of the cervical extensor muscles gradually increases. *Patwardhan et al. (2018)* reported that an increase in T1S can reduce the tension of the posterior ligament complex and cervical extensor muscles. Since the T1 vertebral body acts as the base of the cervical spine, an increase in the T1 tilt angle will cause the base to lean forward, causing the an increase in the C2–C7 SVA (*Le Huec et al., 2019*). The horizontal forward shift of the center of gravity will increase the energy consumption of cervical movement (*Scheer, Lau & Ames, 2021*), stretching the extensor muscles[6] and finally causing a tendency of cervical spine instability. To maintain stability, the active and antagonistic muscles of the cervical spine must then do extra work to form a "muscular collar".

This cross-sectional study has some limitations. First, this study only analyzed cervical muscle function during flexion and extension movement; assessment of lateral bending, rotation, and composite motion are needed in future studies. Sencondly, body builders, and those who have undergone physical therapy, *etc*. Will be incorporated into future studies to control for variables and assess the quantitative impact of these factors on muscular functions. In addition, cross-sectional studies can only indicate whether there is a correlation between variables, a prospective study should be designed to reveal the causal relationship between muscular function and cervical sagittal parameters, in order to determine whether muscular function is an independent influencing factor for cervical alignment.

## CONCLUSIONS

The findings of this study are as follows: (1) Cervical spine imaging parameters differed among subjects with different types of cervical kyphosis. (2) Subjects with R-type kyphosis had worse cervical muscle relaxation and contraction function, and coordination than subjects with other types of kyphosis. (3) The flexion-relaxation and co-contraction functions of cervical muscles were correlated with the imaging parameters of the cervical spine, especially the degree of FHP. (4) Finally, in patients with cervical spine malalignment, co-contraction of the flexor and extensor muscles was observed to form a "muscular collar". The findings of this study can provide a basis for explaining the differences in cervical imaging parameters and muscle functions among subjects with different cervical spine alignments, and provide a reference for symptom assessment and treatment planning.

### Funding

This work was supported by the Beijing Municipal Health Commission, high-level public health technical personnel (2022-3-049). The funders had no role in study design, data collection and analysis, decision to publish, or preparation of the manuscript.

### Grant Disclosures

The following grant information was disclosed by the authors:
Beijing Municipal Health Commission, High-level Public Health Technical Personnel: 2022-3-049.

### Competing Interests

The authors declare that they have no competing interests.

### Author Contributions

- Dian Wang conceived and designed the experiments, performed the experiments, analyzed the data, prepared figures and/or tables, authored or reviewed drafts of the article, and approved the final draft.
- Shuanghe Liu conceived and designed the experiments, authored or reviewed drafts of the article, and approved the final draft.
- Yibo Liu conceived and designed the experiments, authored or reviewed drafts of the article, and approved the final draft.
- Zheng Zeng conceived and designed the experiments, authored or reviewed drafts of the article, and approved the final draft.

### Human Ethics

The following information was supplied relating to ethical approvals (*i.e.*, approving body and any reference numbers):

The ethics committee of our hospital (Beijing Tiantan Hospital, Capital Medical University) approved this study according to the human subject protection programs and procedures (ethics number: KY2021-254-03).

### Data Availability

The raw data are available in the Supplemental File.

### Supplemental Information

Supplemental information for this article can be found online at http://dx.doi.org/10.7717/peerj.18107#supplemental-information.

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
