# Peer review of "Differences in cervical sagittal parameters and muscular function among subjects with different cervical spine alignments: a surface electromyography-based cross-sectional study"

_PeerJ, doi:10.7717/peerj.18107_

## Round 0.1 · original submission · Major Revisions

1. Surface electromyography (sEMG) can be influenced by subcutaneous fat, physiological factors, and the degree of fat infiltration. The study should mention measures taken to account for individual differences during sEMG detection.
2. The limited sample size and lack of control for confounding factors such as physical activity and prior treatments may affect the generalizability and reliability of the results. Future studies should include larger, more diverse samples and account for these confounders to enhance the study's robustness.
3. The authors should consider whether participants with neck pain greater than 3 points should be included in the study.
4. Cervical degenerative changes are likely to be accompanied by disc herniation or spinal stenosis, with symptoms of nerve damage. These patients may be suitable for the study but are not reflected in the inclusion and exclusion criteria.
5. Surface EMG has relatively strict experimental environment requirements, which should be indicated in the article.
6. "Ethics number" is an improper expression. It is suggested to change it to "Ethics review approval number".
7. The cross-sectional study design carries the risk of bias, and the causal relationship between muscle factors, quality of life, and imaging parameters is not yet clear. The authors should acknowledge these limitations and suggest further research to address them.
8. In Table 1, the parameters on the horizontal axis are not aligned in the same row. The table is split across two pages, with inconsistent parameter names (e.g., C2-C7 SVA and CSVA). The alignment of correlation values as positive and negative makes the table data appear disorganized.
9. In Figures 3 and 6, the size and format of each subfigure are inconsistent.

·

Basic reporting

This paper presents an interesting and compelling study. The authors not only categorize cervical kyphosis morphologies but also conduct a qualitative and quantitative evaluation of flexion and extension muscular functions using sEMG in patients with cervical lordosis and kyphosis. The English grammar and writing in this article are impeccable. Hence, I highly recommend considering this paper for publication.

Experimental design

The study demonstrates scientific rigor in its design, and thoroughness in data collection.

Validity of the findings

The statistical methodology utilized in this paper is robust, and the conclusion drawn is precise and reliable.

Additional comments

No additional comments.

Reviewer 2 ·

Basic reporting

Dear Editor and Authors,

Wang Dian et al. conducted a study on the differences in cervical sagittal parameters and muscle function among subjects with different cervical spine alignments using surface electromyography. After careful consideration, I recommend rejecting the manuscript. The primary reasons for rejection are as follows:

Rigorousness: Surface electromyography (sEMG) is used to assess muscle function and activity, which can be influenced by subcutaneous fat, physiological factors of the subjects, and the degree of fat infiltration. The study does not mention any measures taken to account for individual differences during sEMG detection.

Tables and Figures: In Table 1, the parameters on the horizontal axis are not aligned in the same row. Table 1 is split across two pages, with one parameter represented as C2-C7 SVA on the vertical axis and as CSVA on the horizontal axis. The alignment of correlation values as positive and negative makes the table data appear disorganized. In Figures 3 and 6, the size and format of each subfigure are inconsistent.

Reference Format: Some references have formatting inconsistencies or include extraneous symbols such as "<pdf>", for example, in lines 437, 464, 471, 477, 494, 496-498, 518-519, and 525, 531. Some references lack the source journal, such as in lines 502-504, 514-515, etc.

Therefore, I suggest rejecting the manuscript.

Experimental design

no comment

Validity of the findings

no comment

Additional comments

no comment

Reviewer 3 ·

Basic reporting

The study makes a significant contribution to understanding the relationship between cervical sagittal alignment and muscular function.

Experimental design

The study features a clear research question and employs rigorous methodology with detailed experimental procedures, making the findings replicable. However, the limited sample size and lack of control for confounding factors such as physical activity and prior treatments may affect the generalizability and reliability of the results. Future studies should include larger, more diverse samples and account for these confounders to enhance the study’s robustness.

Validity of the findings

Comprehensive data analysis and correlation analysis strengthen the validity of the findings, which are well-aligned with the research questions. Nevertheless, the cross-sectional design limits causal inference, and the generalizability of the results is constrained by the small sample size and unadjusted confounding factors. Acknowledging these limitations and suggesting longitudinal studies and adjustments for confounders in future research would improve the validity and applicability of the findings.

Reviewer 4 ·

Basic reporting

The authors used surface electromyography to evaluate the difference in cervical and shoulder muscle function. The evaluation tool shows better qualitative and quantitative evaluation. Generally, this study is interesting with novelty and clinical significance. I have a few comments.

Experimental design

Inclusion and exclusion criteria. The author should consider that whether the participants with neck pain greater than 3 points should be included in the study. In addition, cervical degenerative changes are likely to be accompanied by disc herniation or spinal stenosis, with symptoms of nerve damage, and these patients are suitable for study. These are not reflected in the inclusion and exclusion criteria.

Validity of the findings

no comment

Additional comments

The authors used surface electromyography to evaluate the difference in cervical and shoulder muscle function. The evaluation tool shows better qualitative and quantitative evaluation. Generally, this study is interesting with novelty and clinical significance. I have a few comments.

1. Inclusion and exclusion criteria. The author should consider that whether the participants with neck pain greater than 3 points should be included in the study. In addition, cervical degenerative changes are likely to be accompanied by disc herniation or spinal stenosis, with symptoms of nerve damage, and these patients are suitable for study. These are not reflected in the inclusion and exclusion criteria.

2. Surface EMG has relatively strict experimental environment requirements, which should be indicated in the article

3. "ethics number", improper expression. It is suggested to change to " Ethics review approval number".

4. Some references are not written in standard, so it is recommended to carefully correct errors and improve them.

5. The paper fails to make clear suggestions on the future research direction, and I hope it can be amended.

6. The study design is a cross-sectional study, with the risk of bias, and the causal relationship between muscle factors, quality of life and imaging parameters is not yet clear, suggesting further research in the future.

---

## Round 0.2 · accepted · Accept

Since all comments have been fully addressed by authors, I think this paper can be accepted for publication.

Reviewer 3 ·

Basic reporting

No more comments

Experimental design

No more comments

Validity of the findings

No more comments

Additional comments

No more comments

Reviewer 4 ·

Basic reporting

The authors have revised following my suggestions. I have no further comments.

Experimental design

No comment.

Validity of the findings

No comment.

Additional comments

No comment.